# Rapid Estimation of Crop Water Stress Index on Tomato Growth

**DOI:** 10.3390/s21155142

**Published:** 2021-07-29

**Authors:** Kelvin Edom Alordzinu, Jiuhao Li, Yubin Lan, Sadick Amoakohene Appiah, Alaa AL Aasmi, Hao Wang

**Affiliations:** 1College of Water Conservancy and Civil Engineering, South China Agriculture University, No. 483, Wushan Road, Tianhe District, Guangzhou 510642, China; kelvinedomalordzinu@gmail.com (K.E.A.); aasadick07@gmail.com (S.A.A.); alaaasmi83@gmail.com (A.A.A.); whao20000904@gmail.com (H.W.); 2College of Engineering, National Center for International Collaboration Research on Precision Agricultural Aviation Pesticides Spraying Technology (NPAAC), South China Agriculture University, No. 483, Wushan Road, Tianhe District, Guangzhou 510642, China; ylan@scau.edu.cn

**Keywords:** crop water stress index (CWSI), vegetative water content (VWC), empirical mode, theoretical mode, differential mode, sandy loam and silt loam soils, irrigation scheduling, tomato

## Abstract

The goal of this research is to use a WORKSWELL WIRIS AGRO R INFRARED CAMERA (WWARIC) to assess the crop water stress index (CWSI_W_) on tomato growth in two soil types. This normalized index (CWSI) can map water stress to prevent drought, mapping yield, and irrigation scheduling. The canopy temperature, air temperature, and vapor pressure deficit were measured and used to calculate the empirical value of the CWSI based on the Idso approach (CWSI_Idso_). The vegetation water content (VWC) was also measured at each growth stage of tomato growth. The research was conducted as a 2 × 4 factorial experiment arranged in a Completely Randomized Block Design. The treatments imposed were two soil types: sandy loam and silt loam, with four water stress treatment levels at 70–100% FC, 60–70% FC, 50–60% FC, and 40–50% FC on the growth of tomatoes to assess the water stress. The results revealed that CWSI_Idso_ and CWSI_W_ proved a strong correlation in estimating the crop water status at R^2^ above 0.60 at each growth stage in both soil types. The fruit expansion stage showed the highest correlation at R^2^ = 0.8363 in sandy loam and R^2^ = 0.7611 in silt loam. VWC and CWSI_W_ showed a negative relationship with a strong correlation at all the growth stages with R^2^ values above 0.8 at *p* < 0.05 in both soil types. Similarly, the CWSI_W_ and yield also showed a negative relationship and a strong correlation with R^2^ values above 0.95, which indicated that increasing the CWSI_W_ had a negative effect on the yield. However, the total marketable yield ranged from 2.02 to 6.8 kg plant^−1^ in sandy loam soil and 1.75 to 5.4 kg plant^−1^ in silty loam soil from a low to high CWSI_W_. The highest mean marketable yield was obtained in sandy loam soil at 70–100% FC (0.0 < CWSI_W_ ≤ 0.25), while the least-marketable yield was obtained in silty loam soil 40–50% FC (0.75 < CWSI_W_ ≤ 1.0); hence, it is ideal for maintaining the crop water status between 0.0 < CWSI_W_ ≤ 0.25 for the optimum yield. These experimental results proved that the WWARIC effectively assesses the crop water stress index (CWSI_W_) in tomatoes for mapping the yield and irrigation scheduling.

## 1. Introduction

Water insufficiency resulting from climate change, global warming, and the competition of freshwater for irrigation with domestic and industrial users is limiting agricultural production. Drought is causing a global fear and threat to food and water security, hence the need to efficiently improve the water use efficiency by adopting good water management practices, irrigation scheduling, and concentrating on plant water status identification. Soil water deficit is a significant aspect of irrigation and soil management that contributes to water stress in plants, hence negatively affecting the crop productivity [1,2]. The estimation of crop water stress and utilizing appropriate irrigation scheduling methods based on soil type and crop type at each stage of the crop is very vital in crop yield optimization [3,4]. However, the timely and accurate identification of the crop water status is essential in making good irrigation scheduling decisions. A reduction in soil water (increasing the soil water deficit) causes stress in a crop growth due to its negative effects on the plant’s biophysical and biochemical properties. It is, however, important to estimate the water status in a plant during each developmental stage in different soil % FC for effective irrigation scheduling [5]. Generally, the in-situ measurement of the crop water status and soil moisture content is laborious, time-consuming, and destructive [6,7,8], hence the need for a timely, accurate, easy, and nondestructive method of detecting the water status in crops early enough to curb damage to the yield and economic losses. According to Osroosh et al., 2015 [9], effective water savings may augment agricultural sustainability through appropriate irrigation scheduling methods. Recently, most researchers have adopted remote sensing data to substitute the old-style field measurements of the plant water status for estimating crop water stress. This provides information on the spatial and temporal variabilities of crops. According to Ihuoma and Madramootoo (2019), Nemeskéri et al., 2015, Ustin et al., 2004, and Zarco-Tejada and Ustin, 2001 [10,11,12,13], estimating the water status in plants is a suitable tool in scheduling irrigation and making water management decisions. However, the temperature approach has been used effectively in remote sensing to estimate the crop water stress index (CWSI) since the 1980s (Idso (1982) and Jackson (1982) [14,15], because the crop canopy temperature can be substituted for crop water stress detection. The crop water stress index (CWSI) was designed as a standardized water index, because it is a fast, easy, and accurate technique for estimating the crop water status over most VIs. Additionally, many researchers in remote sensing (RS) for precision agriculture technology (PAT) have used different approaches and data acquisition tools for estimating the CWSI in a crop canopy. Idso et al., 1981 [16] used the empirical method to estimate the CWSI. Veysi et al., 2017 [17] used satellite infrared reflectance information obtained from satellite images via hot and cold pixels to estimate the CWSI on sugarcane fields. The electromagnetic reflectance obtained in the visible light and near-, mid-, and far-infrared spectral regions from plant leaves was used in computing and mapping the leaf temperatures used to estimate the CWSI in water-stressed plants [18,19]. Other researchers have also used UAV and satellite-based and ground-based hyperspectral and multispectral data for estimating the water stress in crops. However, Matese et al., 2018 [20] reported that the RS information obtained from a plant canopy by a high-resolution thermal infrared camera has numerous spatial and temporal resolutions at a few distances above the plant. This can provide timely and essential information for estimating the crop water status. According to Chen et al., 2005, Jackson, 1982, and Jackson et al., 2004 [14,21,22], the CWSI being a standardized indicator of water stress in soils and plants based on thermal reflectance have frequently been utilized to evaluate the water status in countless numbers of vegetations from forest plants to vegetable plants as in grapevines [20], wheat [23,24,25], tomatoes (Zhang et al., 2017) [1], maize (Zhang and Zhou 2017, Taghvaeian et al., 2012, and Zhang et al., 2019) [26,27,28] and cotton [12,29]. However, this methods of calculating the CWSI for estimating water stress in crops requires more time and high ICT intellectuality, hence creating a big gap between researchers and farmers. To bridge this gap and off-set these pending problems, the utilization of a novel CWSI camera that directly estimates the CWSI devoid of secondary weather forecast data, multifaceted mathematical formulae, and software, hence the need for a Workswell WIRIS AGRO R INFRARED CAMERA (WWARIC). The WWARIC device was intended to map the crop stress in a wide range of agricultural production fields, regardless of crop type and environmental conditions. This device was designed purposely for CWSI estimation, and it can be mounted on UAV, ground-based, and satellite remote sensing platforms. The device is vital in precision agriculture in facilitating on-farm water stress estimations in all weather conditions at different plant populations and developmental stages. This piece of evidence is momentously utilized in yield estimations, enhancing irrigations and fine-tuning water supervision strategies. This study aimed at using a Workswell WIRIS AGRO R INFRARED CAMERA (WWARIC) to estimate the water stress in tomato growth in sandy loam and silt loam soils.

## 2. Materials and Methods

Pot trials were piloted in a greenhouse at the Guangdong Academy of Sciences, Tea Research Institute, South China Agricultural University, Guangzhou, China, 2020 to 2021. The research site is located between Latitude 23°157′826″ N and Longitude 113°350′668″ E, with an elevation of 30 m. The experimental area was a 40 m^2^ field. A 2 × 4 (two soil texture types * four water stress levels) factorial experimental design was set up in a randomized complete block design (RCBD) replicated four times. The water stress treatments applied ranged from 70% to 100%, 60% to 70%, 50% to 60%, and 40% to 50% FC with each replicated four times to give a total of 32 plants. Tomato (Solanum Lycopersicum) cv. Xiang Sheng seeds were nursed in a 6 × 12 nursery tray in cocopeat on the 15 October 2020 and transplanted on the 10 November 2020 into truncated black plastic pots (single plant per pot) with 1 × 0.4-m plant spacing within a block and 1.5 m between block spacing, with black, woven black rubber mulch that was used as a greenhouse floor cover to control weeds. 

### 2.1. Soil Data

Sandy and silty loam soils where the two types used in this experiment. These soil types were collected from the South China Agricultural University Crop research farm. The physical and chemical properties of the soil were analyzed using standardized methods. The results of these properties are shown in Table 1 and Table 2.

### 2.2. Water Stress Treatment

Water through a gravity-driven trickle irrigation system with an hourly flow rate of 1.2 L was estimated and standardized in the greenhouse at field conditions. The water stress treatment applied was built on the soil field capacity. Time Domain Reflectometers (TDR) was used to sense the volume of water in each treatment pot at the different developmental stages of tomato growth. The water stress treatment levels for each treatment used were 70–100% FC, 60–70% FC, 50–60% FC, and 40–50% FC. The upper and lower irrigation thresholds were set at a 10% diminution of soil FC% in all the applied treatments in both soils. The volumetric soil water content and the Time Domain Reflectometry (IMKO. TRIME. PICO TDR HD2 64) water readings conducted every 12 h before and after irrigating the plants were used as a baseline before water application. TDR was implanted at a 30-cm depth in the soil, approximately the root depth of the tomato. The water stress treatment was introduced when the plants developed six true leaves. The volume of water applied to soil to bring the soil back to field capacity was calculated based on the resulting equation by Hamouda et al., 2019 [2]: (1)$$I=\frac{Q+\left(SWC-AWC\right)\times DSM}{A}\times 1000$$
where *I* denotes irrigation water (mm), *Q* the volume of ponding water (mm), *SWC* is the saturated water content of the soil (%), and *AWC* is the actual water of the soil when irrigating (%).

High-quality irrigation water was applied through a drip system, with emitters placed in each pot. 

The TDR and the volumetric techniques measured different values of soil water contents for the same soil type at the same field capacity. A coefficient of determining the correlation R^2^ was used to establish a relation between the TDR readings and the gravimetric reading to reflect the Available Soil Water Content (AWC) for the two soils used for the experiment to maintain a continuous and constant FC% for each treatment. The comparison of the TDR probe moisture values and gravimetric moisture values plotted against each other to report the correlation amid the TDR probe moisture values and VWC values estimated in the laboratory. The volumetric water content was calculated using Equation (2):(2)$$\theta =\frac{W\rho b}{\rho w}$$
where *θ* (cm^3^cm^−3^) is the volumetric water content, *w* (g/g) is the gravimetric water content, **ρ***b* (gcm^−3^) is the soil density, and **ρ***w* (gcm^−3^) is the density of the water.

### 2.3. Crop Water Stress Index (CWSI_W_)

The Crop Water Stress Index (CWSI_W_) was gained by using a WORKSWELL WIRIS AGRO R INFRARED CAMERA (WWARIC). Figure 1 shows the image of the WWARIC. This is an onboard processing camera for evaluating the Crop Water Stress Index (CWSI) and temperature. The operation of WIRIS OS for real-time information streaming and evaluations during the flight has an operating system that ensures full access to all camera functions. The camera can be easily controlled via S. Bus, CAN bus, MavLink, RJ-45, or Trigger. This camera’s plant biomass coverage catalogue is the real-time estimation of vegetation mass in the RGB image format. WWARIC has a sensor resolution-specific at 640 × 512 pixels. The CWSI evaluated from WWARIC has a standardized value ranging from 0 to 1, indicating the degree of stress in the plant to the pixel value. This data can be used to forecast the crop yield and make sound water management policies in precision agriculture. The FPA vigorous sensor scope of WWARIC is 1088 × 10^−3^ × 8705 × 10^−3^ cm with an LWIR band sensor ranging from 0% to 100% (100% is very stressed towards 0, which means less stress) with a temperature compassion of 0.03 °C (30 mK). The field of view of WWARIC is lens 45°; it has 4 color maps, including CROP palettes, CROP STEP palettes, WATER palettes, and WATER STEP palettes of the CWSI. It has programmed settings with a digital zoom of 1–14x continuous. The CWSI camera has software called CorePlayer; the package includes two licenses and 3D mapping. The SW is compatible with Agrisoft and Pix4D, 10× optical zoom RGB Resolution 1920 × 1080 Full HD 1/3 sensor, white auto stability, varied range, backlight recompence, exposure, and gamma control system. WWARIC has 10× optical zoom with tremor recompence and an angle of ultra-zoom 6.9° that can be zoomed to 58.2° and a focal length of 33.0 mm–3.3 mm. The Auto-focus of WWARIC is direct. It has a high-speed SSD of 128 GB for image and video streaming and an External SD card and USB 2.0 slot image storage. The CWSI image from WWARIC is an electromagnetic reflectance that can be stored as JPEG images, TIFF, and Digital camera FullHD JPEG images and in a digital raw data recording.

### 2.4. Camera Measurement Functions

The WWARIC for estimating the CWSI has four-color maps with full radiometric (temperature) information. The real-time intermittent image acquisition includes a digital CWSI image, CWSI video, and visible image concurrently. The camera modes are conception images, full-screen RGB with a subdivision that has a dual-screen micro-HDMI video output 1280 × 720 pixels (720p), and a feature ratio of 16:9 with HDMI and SDK video output software for computers. The Innovative Workswell CorePlayer for offline CWSI image analysis inputs a power source ranging from 9 to 36 DCV and a co-axial of 2 × 6.4 mm with 12 watts in the outmost cover GND Power degeneracy. The WWARIC has an average mass of 430 g with 83 mm in length, 85 mm in width, and 68 mm in height. This camera can be mounted onto a 2 × 1/4-20 UNC support system coupled with a tripod stand, UAV, and satellite system. The camera case is made up of a rigid aluminum plate to withstand external heat and protect the internal system from damage. It can be operated in environmental temperatures (lowest temperature of −10 °C and highest temperature of 50 °C and can be well-stored at a minimum temperature of 30 °C and maximum temperature of 60 °C. 

### 2.5. Field Measurements

#### 2.5.1. Image Acquisition

The WORKSWELL WIRIS AGRO R INFRARED CWSI CAMERA was made in the Czech Republic, version 1.38, resolution 640 × 512, RTC temperature 37.6 °C, CPU temperature 46.3 °C, and IR core temperature at 45.8 °C mounted at the height of 4 m and used to acquire images on tomato crops grown under different soil field capacities to estimate the crop water status in tomato growth. Data obtained from the CWSI camera was analyzed with CorePlayer software version 1.3.8. CWSI_W_ was compared with CWSI_Idso_ and the vegetative water content VWC to determine the accuracy through a correlation.

#### 2.5.2. Canopy Temperature Measurement

The canopy temperature was estimated using a Raytek^®^ portable, noncontact infrared thermometer with an emissivity of 0.95 Wm^−2^ and a display resolution of 0.2 °C (0.5 °F). The thermometer was held 30 cm overhead the tomato leaves, with the optical maser point set at an angle of 90° perpendicular to the target leaves [10,30]. Temperature readings were taken at each growth stage of the tomato cycle on the five sampled plants from individual treatments. Leaf temperatures were taken at an all-round direction following the cardinal points starting from the north, and the mean temperature for each leaf in each treatment was calculated. Temperature readings were taken at 11:00 a.m. and 2:00 p.m. on a bright, sunny day; with sunlight in the greenhouse on the fully developed leaves at the plant apex, as recommended by Ihuoma and Madramootoo (2019) [10].

#### 2.5.3. The Conceptualization of CWSI Founded on the Idso Method

Idso estimated the CWSI empirically based on the linear relationship observed as the difference in the crop canopy temperature to air temperature (Tc−Ta) and vapor pressure deficit (VPD) of the air in the good irrigated plant (Idso et al., 1981 and Veysi et al., 2017) [16,17] in a bright sunny day under standardized environment conditions. To estimate the CWSI, two reference points must be determined (lower reference point and upper reference point), which must be crop-specific (Idso 1982, Jackson et al., 1981, and Katsoulas et al., 2016) [15,31,32]. According to Idso et al., 1981 and Jackson et al., 1981 [16,31], the lower reference point represents the no water-stressed level that is dependent on the vapor pressure deficit (VPD), whilst the upper reference point represents a water-stressed level. It is based on the crop canopy without transpiration (Tc−Ta); this is not dependent on the vapor pressure deficit (VPD). However, the CWSI_Idso_ is then calculated based on the equation formulated by (Idso et al., 1981) [16]:(3)$${\mathrm{CWSI}}_{\mathrm{IDSO}}=\frac{(T_{C}-T_{a})-\left(T_{C}-T_{a}\right)\;\mathrm{UL}}{(T_{C}-T_{a})\;\mathrm{UL}-\left(T_{C}-T_{a}\;\right)\mathrm{LL}}$$

Tc represents the tomato leaf cover temperature (°C), Ta represents the greenhouse air temperature (°C), and LL signifies the lower reference point where plants have an adequate amount of water, and UL signifies the upper reference point where the plant is highly water-stressed. The lower reference point for the crop covering-to-air temperature difference (Tc–Ta) against the vapor pressure deficit (VPD) association was estimated from the information gathered solely on the water stress treatment at different developmental stages of tomato growths. To confirm the upper reference point, the canopy temperatures of completely stressed crops were assessed daily during tomato growth (vegetative growth, anthesis stage, fruit enlargement stage, and senescence stage. Vital weather conditions that were used to estimate the CWSI_Idso_ are shown in Table 3. This index defines the CROP water status on a scale ranging from 0 to 1. 0 signifies no or very low water stress, and 1 signifies a high-water stress

#### 2.5.4. Crop Sampling and VWC Calculation

The vegetation water content (VWC) was estimated at all the developmental stages of tomato growth by sampling young and fully mature leaves at the plant apexes from 4 plants in all the treatments. The picked leaves were enclosed in a rubber zip bag, reserved in a cooling chamber at five °C, and transported to the laboratory. A fresh leaf mass (FW) was obtained utilizing an electronic scale, and the values were recorded. Leaves were oven-dried at a temperature of 72 °C to a constant dry weight (DW). Table 4 shows the mean VWC at all the developmental stages in the tomato growth, and this was calculated using the equation by reference [17]:(4)$$\mathrm{VWC}=\frac{\left(\mathrm{FM}-\mathrm{DM}\right)}{\mathrm{DM}}\times 100$$

#### 2.5.5. CWSI_W_ Estimation

The CWSI infrared camera has three modes that can estimate the crop water stress index (CWSI) in the manual temperature range. This menu consists of an empirical mode with stress levels between 0% and 100%. A 100% stress level is selected when the central cross point is very dry. In contrast, the 0% stress level is selected when the area is thoroughly watered. The Differential mode (DIFF) uses only the air temperature, i.e., current air temperature, to estimate the CWSI_W_. The third model, the theoretical model, uses the crop type, air temperature (°C), relative humidity (%), the slope of the baseline, and the baseline intercept to estimate the CWSI in the crop. 

## 3. Results 

### 3.1. Soil Water Content Measurement

The results from this study revealed that the TDR and gravimetric soil water content measurement had a strong correlation. The correlation coefficient turned out to be greater than 0.90 at *p* < 0.05 for sandy loam and silty loam soils correspondingly, as shown in Figure 2. The total irrigation water applied for tomato growth throughout the growing season was 225.9 mm, 146.9 mm, 124.3 mm, and 101.7 mm at 70–100% FC, 60–70% FC, 50–60% FC, and 40–50% FC for sandy loam soil, respectively, whereas the total amount of irrigation water applied at each treatment in silt loam soils was 243 mm, 158.1 mm, 121.3 mm, and 109.5 mm for 70–100% FC, 60–70% FC, 50–60% FC, and 40–50% FC, respectively. 

#### Soil Moisture Variation

The volumetric water content varied throughout tomato growths at different soil FC in sandy loam and silty loam soils. The results revealed that a water application was frequent throughout the plant’s growth for all the water stress treatment levels throughout the anthesis and fruit development phase. However, these are the stages in the crop’s growth in which the biophysical, biochemical, and physiological developments are perilous and necessitate more water to optimize the yield. The day-to-day volumetric soil water content for various water stress treatments for both soils is shown in Figure 3.

### 3.2. Crop Water Stress Index (CWSI) and Baseline Equations for Tomatoes

The CWSI varied under different water stress treatments in sandy loam and silt loam soils during the tomato growing season; this is shown in Figure 4. The upper reference point limit (URL) and lower reference point limit (LRL) equations were established using these equations UBL = 0.16 VPD + 2.92 and LBL = −1.03 VPD − 2.352 for sandy loam soils and UBL = 0.11 VPD + 4.92 and LBL = −3.03 VPD − 0.352 for silt loam soils similar to the equation used by reference [33]. The CWSI values increased with the increasing water stress. In sandy loam soils, the CWSI values ranged between 0.01 to 0.38 for 70–100% FC, 0.17 to 0.65 for 60–70% FC, 0.47 to 1.09 for 50–60% FC, and 0.51 to 1.22 for 40–50% FC water stress treatments, whilst the silt loam soil CWSI value ranged from 0.02 to 0.35 for 70–100% FC, 0.22 to 0.60 for 60–70% FC, 0.41 to 0.9 for 50–60% FC, and 0.49 to 1.0% FC for 40–50% FC water stress treatments for all the tomato growths.

### 3.3. CWSI Based on the Field Measurement by Idso Method and a WORKSWELL WIRIS AGRO R CAMERA

Tomato CWSI images, digital images, a 3D graph, and an isothermal graph obtained from the WORKSWELL WIRIS AGRO R CAMERA analyzed with WIRIS CORE PLAYER VERSION 1.3.8 are shown in Figure 5, Figure 6, Figure 7 and Figure 8. Figure 9 shows that the results from this experiment revealed a strong correlation between the calculated CWSI_I_ utilizing field measurements from canopy temperatures obtained with a Raytek^®^ handheld, noncontact infrared thermometer, and CWSI estimation using the WORKSWELL WIRIS AGRO R INFRARED CAMERA at each growth stage of tomato growth, with a R^2^ above 0.60 at each growth stage in both sandy loam and silt loam soils. The coefficient of determining (R^2^) at each growth stage was slightly high at the fruit expansion stage at R^2^ = 0.8363 and 0.7611 in sandy loam and silt loam, respectively. This can be associated with the fruit development stage, where the plant utilizes a lot of water. A correlation is somewhat increased in the vegetative stage at R^2^ = 0.746 and 0.7072 for sandy loam and silt loam soils, respectively; this may also result from the reflection soil surface at the early growth stages of the tomato plant. According to Magney et al., 2016 and Sakamoto et al., 2013 [34,35], a minimal vegetation layer produces a high percentage of diverse pixels in a lot of obtainable remote sensing results. Vegetation reflectance signatures are highly affected by the crop characteristics rather than soil characteristics (Ceccato et al., 2002, Idso 1982, Jackson et al., 1981, Liu et al., 2019, Ustin et al., 2004, and Wang et al., 2015) [4,11,15,23,31,36]. However, these results proved that, in most of the water stress values of tomatoes based on the Raytek^®^ handheld, the noncontact infrared thermometer temperature in the Idso technique; estimating the water stress was not much different from a WORKSWELL WIRIS AGRO R INFRARED CAMERA. 

### 3.4. Relationship between CWSI Estimated Using WORKSWELL WIRIS AGRO R INFRARED CAMERA and VWC

The results from this study revealed that a CWSI_W_ estimates using WWARIC and VWC had a strong positive correlation with the R^2^ value equal to 0.8814 and 0.8024, *p* < 0.05 at the various growth stages of tomato growth in sandy loam and silty loam soils, respectively, as shown in Figure 10. This study aims at evaluating the likelihood of using a WORKSWELL WIRIS AGRO R INFRARED CAMERA to accurately predict the CWSI for irrigation scheduling in tomatoes grown in different soil types at different water stress levels under greenhouse conditions. The relationship between measured the VWC and estimated CWSI-based WORKSWELL WIRIS AGRO R INFRARED CAMERA at different growth stages of tomatoes growth in sandy loam and silt loam soils in the greenhouse revealed a negative relation. The results showed that, as the VWC increases, the CWSI_W_ decreases. However, as the tomato plant reaches senescence, the water absorption rate reduces, thereby reducing the transpiration rate and photosynthesis, even though there is enough available soil in the water. In general, the CWSI is a normalized index primarily for estimating the crop water status. The strong correlation obtained between the VWC and estimated CWSI_W_-based WORKSWELL WIRIS AGRO R INFRARED CAMERA CWSI was because both were measured at the leaf/canopy level. Therefore, it is advisable to estimate the correlation of two or more indices on the same plant part to avoid variations. Furthermore, the field measurements of VWC and estimated CWSI-based WORKSWELL WIRIS AGRO R INFRARED CAMERA were significantly different between the water stress levels but not significantly different among the soil types; thus, there was a linear increase in VWC in tomato plants from low water stress to high water stress and vias versa for CWSI for both soil types. When the soil water deficit increases, the crop water status decreases, hence increasing the CWSI index value but decreasing VWC. However, the estimated CWSI-based WORKSWELL WIRIS AGRO R INFRARED CAMERA could be used to effectively evaluate and map the crop water status and make irrigation scheduling management decisions. The results obtained from the estimated CWSI-based WORKSWELL WIRIS AGRO R INFRARED CAMERA of four water stress levels were established for both soil types for all growth stages by dividing the CWSI into a range of 0.25 intervals similar to the approach by references [17,33]. The CWSI at all growth stages at the 70–100% FC water stress treatment for both soil types has a CWSI interval range of 0.0 < CWSI_W_ ≤ 0.25, which is indicated by a deep green color, 60–70% FC water stress treatment of 0.25 < CWSI_W_ ≤ 0.5 indicated by a light green color, 50–60% FC water stress treatment of 0.5 < CWSI_W_ ≤ 0.75 indicated by a yellow color, and 40–50% FC water stress treatment of 0.75 < CWSI_W_ ≤ 1.0 indicated by a light brown color this is represented in Figure 6. However, −1 < CWSI_W_ ≤ 0) and CWSI > 1 represented by blue and deep brown colorations on the tiff images, which indicates excess water and no water availability, respectively. Based on the linear correlation between the estimated CWSI_W_-based WORKSWELL WIRIS AGRO R INFRARED CAMERA and VWC. VWC was classified based on the water stress levels imposed as treatments on tomato growths in both soil types (sandy loam and silt loam soils) in a range of intervals, revealing high-to-low water stress levels. This classification includes 0.65 < CWSI ≤ 0, which is essential for mapping water stress and yields in tomato growths for all the CWSI modes of the WORKSWELL WIRIS AGRO R INFRARED CAMERA. The mapped area by the color mapping is shown in Figure 5.

### 3.5. Correlating CWSI_W_ with Yield 

Figure 11 revealed a strong bond between the yield and estimated mean CWSI_W_ in sandy loam and silt loam soils. There is a strong positive correlation between the CWSI_W_ and yield at R^2^ = 09948 and 09813 for sandy loam and silt loam soils, respectively. The study also revealed that the yield increased with the decreasing CWSI_W_; this is because, at a high soil water deficit, crop water uptakes are reduced, thereby reducing the transpiration rates and increasing the crop canopy temperatures, leading to yield reduction and economic loss. The CWSI_W_ has proven a vital tool for mapping crop yields.

## 4. Discussions

CWSI Indices have been utilized in most agricultural sectors, especially in the field of soil and crop science for the estimation of water stress or soil water deficits. However, due to the complex nature, high technology, and the time frame involved in estimating the water status to effectively curb plant mortality and yield loss, most researchers in the field of precision agriculture technology have employed remote sensing coupled with high-resolution cameras to solve this problem (Idso et al., 1981, Jackson et al., 1981, Sepaskhah and Kashefipour 1994, and Tanriverdi et al., 2017) [16,31,37,38]. The CWSI was first calculated by Idso in the early 1980s using air temperature and vapor pressure deficits. Aforementioned researches have proven that there was a high correlation between the CWSI estimated by Idso, the CWSI estimated by other methods, and the VWC obtained from field data (Cao et al., 2015, Espinoza et al., 2017, Trenberth et al., 2015, and Zhang et al., 2017). Gerhards et al., 2019 [1,30,39,40,41] used TIR remote sensing data to estimate the CWSI, which revealed a strong positive correlation with the Idso method of estimating the CWSI_IDSO_ with a R^2^ above 0.9. The same can be said of Poblete-Echeverría et al., 2017 [42], who estimated the CWSI using the empirical and theoretical methods. The results from this study in Figure 2 show that there is a strong correlation between the TDR soil moisture readings and the gravimetric moisture estimation at R^2^ above 0.9 in sandy loam and silt loam soils; hence, the TDR has the potential to accurately estimate the soil water content in all soil types; this finding is comparable to the findings of references [43,44,45,46,47,48,49], who used the TDR to estimate the soil water deficit for the estimation of water stress in a crop canopy by establishing the relationship between the TDR and gravimetric soil moisture readings. Similarly, based on the baseline equation of tomatoes established in estimating the CWSI, it showed that the CWSI_w_ and CWSI_IDSO_ values increased with the increasing water stress in sandy loam and silt loam soils. The CWSI values obtained in the study on tomato growth in sandy loam soil ranged between 0.01 and 0.38 for 70–100% FC, 0.17 and 0.65 for 60–70% FC, 0.47 and 1.0 for 50–60% FC, and 0.51 and 1.0 for 40–50% FC of the water stress treatments, whilst, in silt loam soil, the CWSI values ranged from 0.02 to 0.35 for 70–100% FC, 0.22 to 0.60 for 60–70% FC, 0.41 to 0.9 for 50–60% FC, and 0.49 to 1.0% FC for 40–50% FC for the water stress treatments throughout the growing season; these results are similar to the findings of Harmanto et al., 2005, Jackson et al., 1981, Mohamed and EL-Aziz 2020, Pék et al., 2014, Sepaskhah and Kashefipour 1994, Silber et al., 2015, Tanriverdi et al., 2017, Veysi et al., 2017, and Yang et al., 2014) [3,17,31,37,38,50,51,52,53], who reported that the CWSI increases with the increasing soil water deficit. The crop water stress index (CWSI_IDSO_) obtained from field measurements by the Idso method and the WORKSWELL WIRIS AGRO R INFRARED CAMERA also showed a good correlation at a R^2^ above 0.9, proposing that the WORKSWELL WIRIS AGRO R INFRARED CAMERA has the feasibility of estimating the CWSI index for evaluating a timely and accurately crop water status. However, the modes of CWSI in the WORKSWELL WIRIS AGRO R INFRARED CAMERA for mapping crop water stress—that is, the empirical model and theoretical mode—proved excellent color mapping for the CWSI_W_ estimation within a height of 0–5 m above the crop canopy, as shown in Figure 5, whilst the differential mode showed an improvement in estimating the CWSI_W_ at a height less than 10 m and was suitable to estimate the CWSI_W_ index at a higher height somewhat between 10 m and 30 m or above the crop canopy; hence, this mode will be helpful when using an UAV or satellite. The results obtained from the empirical and the theoretical modes from the WORKSWELL WIRIS AGRO R INFRARED CAMERA are similar to the findings of Gebregziabher et al., 2009, Hussain and Hanjra 2004, Idso 1982, Idso et al., 1981, Jackson et al., 1981, and Katsoulas et al., 2016) [15,16,31,32,54,55], who used field measurement and satellite approach deploying the empirical method to estimate the CWSI index for assessing crop water stress in various crops. The CWSI estimated from the WORKSWELL WIRIS AGRO R INFRARED CAMERA can notice the crop water status instantaneously and accurately. According to Idso et al., 1981, Jackson et al., 1981, and Wang et al., 2011 [16,21,56], the CWSI_IDSO_ is a method that utilizes the vapor pressure deficit (VPD) and crop surface temperature (meteorological data) to estimate the crop water status. However, the WORKSWELL WIRIS AGRO R INFRARED CAMERA can estimate the CWSI_W_ without requiring meteorological data as a secondary data source. Many researchers have used the CWSI index to effectively and successfully estimate the crop water status in various crops after Idso et al., 1981 [16] proposed the CWSI_IDSO_ using the empirical approach. The results in Figure 7 showed that the CWSI_W_ obtained from WORKSWELL WIRIS AGRO R INFRARED CAMERA and VWC was strongly correlated at R^2^ = 0.8814 and 0.8024 for the sandy loam and silt loam soils, respectively, at all growth stages of tomato growth at different water stress levels (70–100% FC, 60–70% FC, 50–60% FC, and 40–50% FC), even though they had a negative relationship. These results are similar to the findings of Jones, 2012; Katsoulas et al., 2016; Lisar et al., 2012; Matese et al., 2018; Tanriverdi et al., 2017; Veysi et al., 2017; Zhang et al., 2019 [17,19,20,27,32,37,57], who used the CWSI_IDSO_ and vegetative water content for estimating irrigation scheduling. Consequently, the CWSI has a negative effect on tomato yield. Increasing the CWSI decreases the yield of tomatoes and increases economic losses. However, the CWSI_W_ had a good correlation with the yield at R^2^ = 0.9948 and 0.9813 for sandy loam and silt loam soils, respectively; these results are comparable to the works of Degirmenci et al., 2016; Ihuoma & Madramootoo, 2019; Kirnak et al., 2001; Nemeskéri et al., 2019; Sakamoto et al., 2013; Veysi et al., 2017; Zhang et al., 2017 [10,17,35,49,58,59].

## 5. Conclusions 

This study offers an advancement towards applying a WORKSWELL WIRIS AGRO R INFRARED CAMERA in remote sensing to estimate the CWSI for evaluating accurately and timely water stress in tomatoes and to model for irrigation scheduling. The CWSI_W_ estimated in tomato growths at 70–100% FC water stress treatment for both soil types, with a CWSI interval range of 0.0 < CWSI ≤ 0.25, which is indicated by a deep green color, a 60–70% FC water stress treatment of 0.25 < CWSI ≤ 0.5, indicated by a light green color, a 50–60% FC water stress treatment of 0.5 < CWSI ≤ 0.75 indicated by a yellow color, and a 40–50% FC water stress treatment of 0.75 < CWSI ≤ 1.0 indicated by a light brown color. However, −1 < CWSI ≤ 0 and CWSI > 1 represented by blue and deep brown colorations on the tiff images indicate excess water and no water availability, respectively. Due to these results, a novel WORKSWELL WIRIS AGRO R INFRARED CAMERA, a new CWSI camera designed based on environmental and crop data that requires no meteorological data for effectively acquiring digital CWSI, and 3D images at the same time were used for estimating the water stress in tomatoes. In this study, the CWSI index was developed based on three modes of pixels: the empirical, theoretical, and differential modes. The CWSI_W_ obtained from this novel were comparable to the results obtained from the CWSI_I_ and vegetation water content (VWC) at each growth stage of tomato growth and the ROI on the canopy. The results revealed that the empirical and theoretical modes of the novel WORKSWELL WIRIS AGRO R INFRARED CAMERA CWSI used to acquire images at a height of 4 m effectively mapped and estimated the water stress in tomato growths. However, the CWSI_W_ maps can be used to estimate the VWC tomato growths in both sandy loam and silt loam soils. However, it can be concluded that, since the WORKSWELL WIRIS AGRO R INFRARED CAMERA was designed to estimate the CWSI, which is a normalized index developed to quantify stress and overcome the effect of other environmental parameters affecting the relationship between stresses, plant temperature, and air temperature, the CWSI obtained from the WORKSWELL WIRIS AGRO R INFRARED CAMERA can be used to map crop water status to prevent a drought or provide corrective measures for rewatering to prevent crop mortality, improve irrigation scheduling, and yield mapping from optimizing tomato productivity. This novel WORKSWELL WIRIS AGRO R INFRARED CAMERA has proven to be accurate, rapid, and user-friendly in estimating the water stress in tomatoes. It has the ability to estimate the CWSI in a color range and in the figures. This, when employed in a remote sensing platform in precision agriculture, will save farmers and researchers time and money.

## Figures and Tables

**Figure 1 sensors-21-05142-f001:**
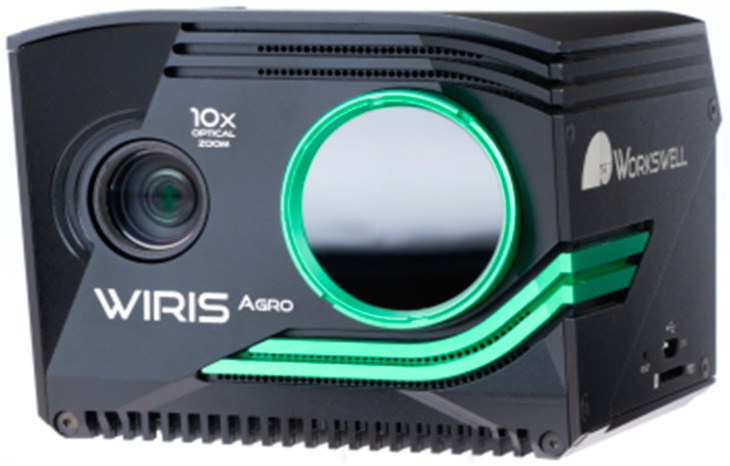
The WORKSWELL WIRIS AGRO R INFRARED CWSI CAMERA.

**Figure 2 sensors-21-05142-f002:**
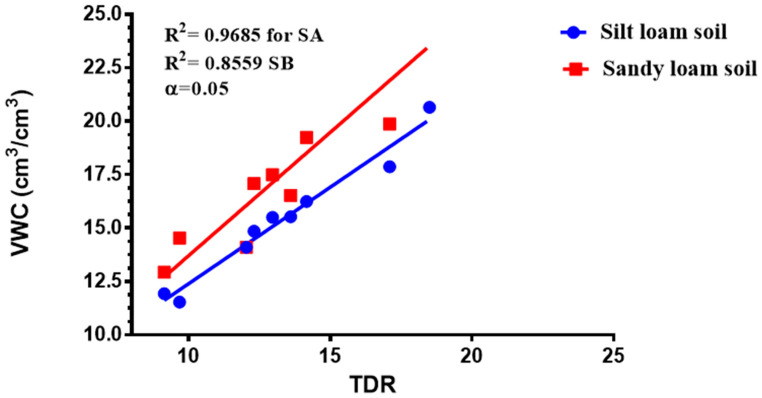
Difference in the moisture content determination by the volumetric water content (VWC) and TDR (±5%) for sandy loam soil (SA) and (SB) silt loam soil.

**Figure 3 sensors-21-05142-f003:**
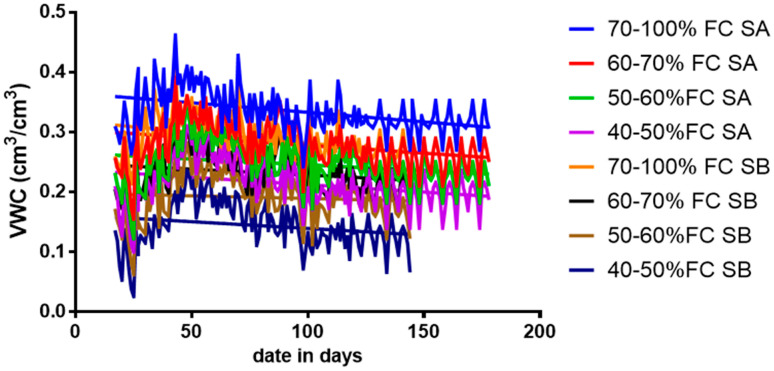
Disparity in the soil moisture content during the growing season for sandy loam soil and silty loam soil.

**Figure 4 sensors-21-05142-f004:**
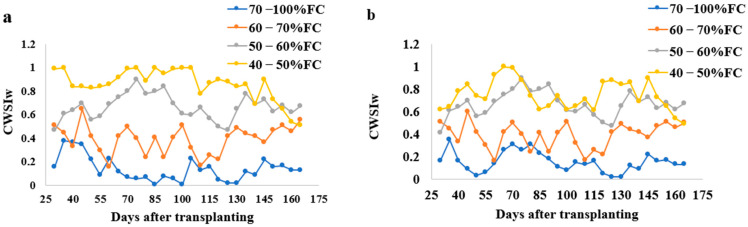
Variations of CWSI_W_ with time during the growing season of tomatoes at different water stress levels in (**a**) sandy loam soil and (**b**) silt loam soil. Note: All the values used in the above figure are the mean values.

**Figure 5 sensors-21-05142-f005:**
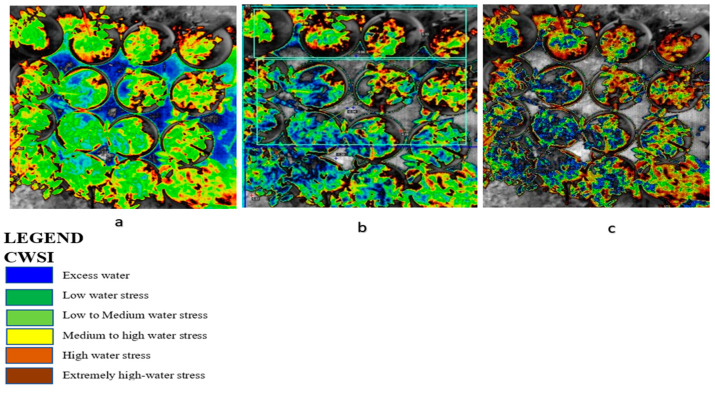
CWSI tiff images from WIRIS CorePlayer at different CWSI modes. (**a**) Empirical mode, (**b**) theoretical mode, and (**c**) differential mode revealing water stress with color mapping in tomato plants.

**Figure 6 sensors-21-05142-f006:**
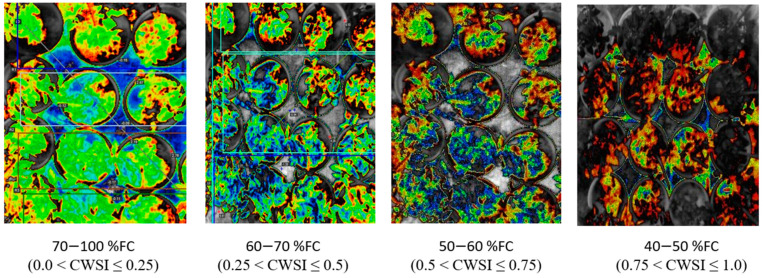
Image colors and corresponding CWSI values at different field capacities in tomato growths.

**Figure 7 sensors-21-05142-f007:**
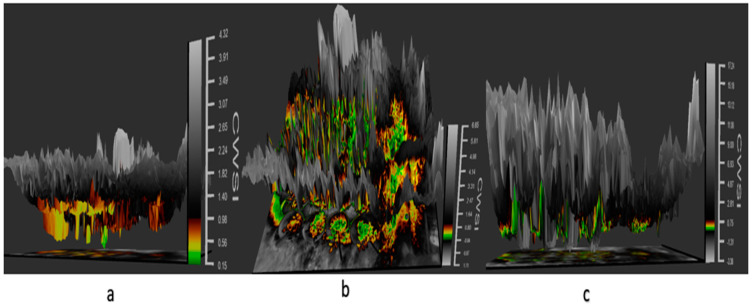
Three-dimensional panel displaying the thermal data in a 3D space of images on a tomato plant in different CWSI modes: (**a**) the differential mode, (**b**) empirical mode, and (**c**) theoretical mode, revealing water stress with color mapping in tomato plants.

**Figure 8 sensors-21-05142-f008:**
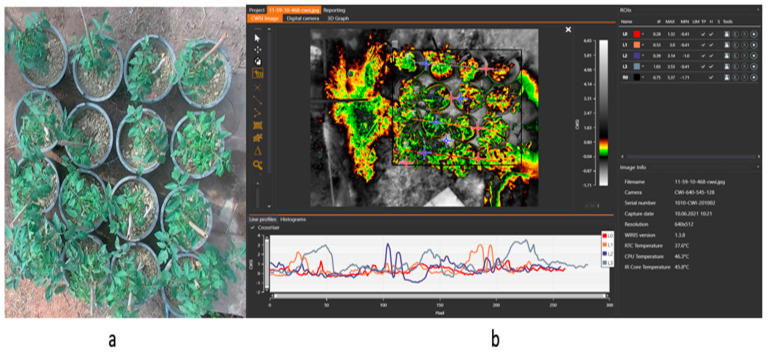
(**a**) Digital image of the tomato plant (**b**) isotherm panel with the map panel and ROI panel reveals the tomato plant’s thermal data for mapping water stress.

**Figure 9 sensors-21-05142-f009:**
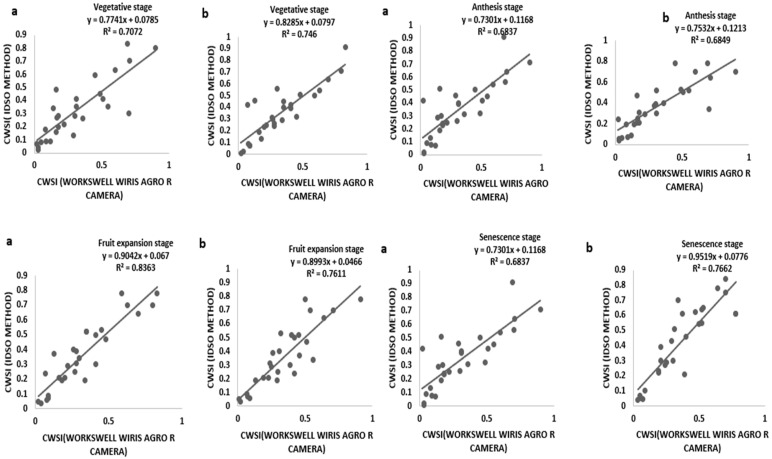
Relationship between the CWSI_IDSO_ and CWSI_W_ ((**a**) sandy loam soil (**b**) silt loam soil) at different growth stages.

**Figure 10 sensors-21-05142-f010:**
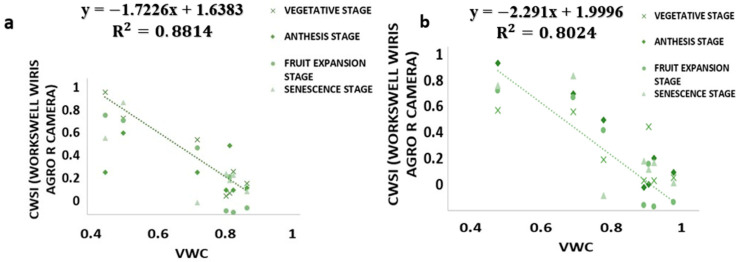
Correlation between the CWSI_W_ and VWC: (**a**) sandy loam soil and (**b**) silt loam soil.

**Figure 11 sensors-21-05142-f011:**
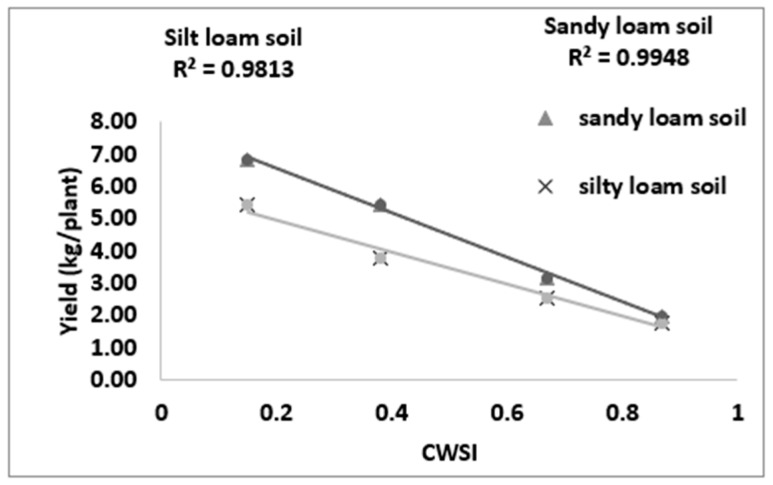
Relationship between the CWSI and yield.

**Table 1 sensors-21-05142-t001:** Results of the soil physical properties.

Soil Texture	Sand (%)	Silt (%)	Clay (%)	Bulk Density (g/cm^3^)	Saturation Point (%)	Field Capacity (%)	Permanent Wilting Point
**Sandy loam**	75.4	20	4.6	1.34	48	21	9
**Silt loam**	43.53	39.93	16.63	1.32	45.73	31	19

**Table 2 sensors-21-05142-t002:** Results of the soil chemical properties.

Soil Texture	pH	O.M (g/kg)	Total N (g/kg)	Total P (g/kg)	Total K (G/KG)	Alkalized N (mg/kg)	Avail. P (mg/kg)	Avail. K (mg/kg)
**Sandy loam**	5.64	15.91	1.23	0.88	9.30	450.28	195.72	428.43
**Silt loam**	5.30	22.97	1.518	0.865	19.59	72.71	28.25	85.50

**Table 3 sensors-21-05142-t003:** Weather conditions at each growth stage.

Growth Stages	Vegetative Stage	Anthesis Stage	Fruit Expansion Stage	Senescence Stage
**Air Temperature (°C)**	30.6	25.1	23.9	27.4
**Relative humidity (%)**	74.6	80.45	70.2	83.2
**Vapor Pressure Deficit**	4.13	5.17	5.02	3.11

Note: Values used are mean values.

**Table 4 sensors-21-05142-t004:** Measured VWC at each growth stage.

Growth Stages	Minimum (%)		Maximum (%)		Mean (%)	
Soil Type	SA	SB	SA	SB	SA	SB
Vegetative Stage	75	69	83	76	79	72.5
Anthesis Stage	80	73	84	80	82	76.5
Friut Expansion Stage	79	82	85	86	82	84
Senescence Stage	73	65	80	74	76.5	69.5

Note: All values in this table are the mean value over the individual growth stages.

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
