# Peer review of "Rapid Estimation of Crop Water Stress Index on Tomato Growth"

_sensors, 2021, doi:10.3390/s21155142_

Round 1

Reviewer 1 Report

The paper presents a method for estimating the influence of “crop water stress index “ on tomato growth. To my understanding and as mentioned in the abstract, the goal of the research is to use a commercial camera (WWARIC) to map water stress. According to the web site of the company selling the camera, the camera is especially designed to map water stress for agriculture applications. It is unclear weather this fits in the scope of this journal.

In the conclusion, the authors mention that the novel camera has been proven to have many advantages and will save farmers time and money. Although this might be true, it is unclear what is the novelty of the paper.

While the study is interesting, it is in my understanding, limited to the usage of a commercial camera to undertake water stress measurements. What is the novelty of the paper?  I believe this study does not fit in the scope of this journal.

Line 317 : the author mentions that figures 3 to 5 present results obtained with the camera. No explanation is provided. The results / figures should be explained in the text.

Author Response

All comments and recommendations have been addressed accordingly.  Please see the attachment

Reviewer 2 Report

The topic is interesting, but the manuscript needs major revision before possible publication. Meanwhile, the language should be improved.

* The title: delete “The study of”, and specify the method.

* Abstract: I cannot see any transition in Line 35-36, why use “however” here?

* Line 109: between?

* Line 110: The experiment should be 2*4 (two soil texture types * four water stress levels) instead of 2*2. More details about the experiment should be given. No soil types mentioned here.

* Line 112: 70-100=>70-100% FC. Similar for others.

* Lines 133-135: was estimated? Built on? Sense? The measurement depth of soil water content should be specified.

* Line 152: duplicate with lines 134-136. Volumetric?

* Line 211: define CWSIIdso first, and use only CWSIIdso or CWSII throughout the manuscript.

* Line 225: Idso => Idso et al. (1981); lined?

* Section 2.5.5: Give the equations to estimate the CWSI.

* Line 259: It was 40-50% FC for sandy loam soil, whereas??

* Line 283: gravimetric or volume?

* Figs. 2 and 3: How was VWC obtained?

* Fig. 4: not clear. Which CWSI? Why was CWSI>1 in some days?

Author Response

All comments and recommendations have been addressed accordingly. Please see the attachment.

Round 2

Reviewer 1 Report

The font size on every figure should be the same (exemple : figure 2 and 3). The font size on every figure should be big enough to be readable.

Reviewer 2 Report

The title covers a very broad topic, and method should be specified.

All abbreviations or variables should be defined in both abstract and main text.

Use only CWSIIdso or CWSII throughout the manuscript, including text and equations.

CWSI is generally in the range of 0-1.
